# Liver Resection for Type IV Perihilar Cholangiocarcinoma: Left or Right Trisectionectomy?

**DOI:** 10.3390/cancers14112791

**Published:** 2022-06-04

**Authors:** Heithem Jeddou, Stylianos Tzedakis, Francesco Orlando, Antoine Robert, Eric Meneyrol, Damien Bergeat, Fabien Robin, Laurent Sulpice, Karim Boudjema

**Affiliations:** 1Department of Hepatobiliary and Digestive Surgery, Pontchaillou Hospital, University of Rennes 1, 35000 Rennes, France; heithem.jeddou@chu-rennes.fr (H.J.); stylianos.tzedakis@aphp.fr (S.T.); francesco.orlando@aocardarelli.it (F.O.); an.robert@chor.re (A.R.); damien.bergeat@chu-rennes.fr (D.B.); fabien.robin@chu-rennes.fr (F.R.); laurent.sulpice@chu-rennes.fr (L.S.); 2Department of Hepatobiliary, Pancreatic, Digestive and Endocrine Surgery, Cochin Hospital, Assistance Publique—Hôpitaux de Paris (APHP), University of Paris, 75014 Paris, France; 3Department of Radiology, Pontchaillou Hospital, University of Rennes 1, 35000 Rennes, France; eric.meneyrol@chu-rennes.fr; 4CIC-INSERM 14-14, University of Rennes 1, 35000 Rennes, France

**Keywords:** perihilar cholangiocarcinoma, Bismuth type IV, right trisectionectomy, left trisectionectomy

## Abstract

**Simple Summary:**

Surgical resection of perihilar cholangiocarcinoma (PHC), a rare malignant bile duct tumor arising in the hepatic hilum, requires either right- or left-sided liver resections depending on tumor-side predominance. Margin-free resection remains the only curative-intent treatment and extended liver resections (trisectionectomies) are needed for advanced (Bismuth type IV) PHC. However, the clinical outcomes of left (LTS) and right trisectionectomies (RTS) have so far not been compared for the resection of advanced PHC. In this retrospective study of consecutive cases of Bismuth type IV PHC, RTS (42 patients) and LTS (25 patients) were compared in terms of postoperative morbidity and patient survival. Although LTS was more frequently associated with arterial reconstructions, the postoperative liver failure rate was lower and overall survival was better as compared to RTS.

**Abstract:**

How the side of an extended liver resection impacts the postoperative prognosis of advanced perihilar cholangiocarcinoma (PHC) is still controversial. We compared the outcomes of right (RTS) and left trisectionectomies (LTS) in Bismuth-Corlette (BC) type IV PHC resection. All patients undergoing RTS or LTS for BC type IV PHC in a single tertiary center between January 2012 and December 2019 were compared retrospectively. The endpoints were perioperative outcomes, long-term overall (OS), and disease-free survival (DFS). Among 67 hepatic resections for BC type IV PHC, 25 (37.3%) were LTS and 42 (63.7%) were RTS. Portal vein and artery resection rates were 40% and 52.4% (*p* = 0.29), and 24% and 0% (*p* < 0.001) in the LTS and RTS groups, respectively. The severe complication (Clavien–Dindo > IIIa) rate was comparable (36% vs. 21.5%, *p* = 0.357) while the postoperative liver failure (POLF) rate was lower in the LTS group (16% vs. 38%, *p* = 0.048). The R0 resection rate was similar between groups (81% vs. 92%; *p* = 0.154). The five-year OS rate was higher in the LTS group (66% vs. 30%, *p* = 0.009) while DFS was comparable (43% vs. 18%, *p* = 0.11). Based on multivariable analysis, the side of the trisectionectomy was an independent predictor of OS. Compared with RTS, LTS is associated with lower POLF and higher overall survival despite more frequent arterial reconstructions in type IV PHC. Although technically more demanding, LTS may be preferred in the treatment of advanced PHC.

## 1. Introduction

Perihilar cholangiocarcinomas (PHC) are rare tumors that arise from the bile duct epithelium of either the extrahepatic main biliary confluence or intrahepatic small bile ducts adjacent to the biliary confluence and invading the hepatic hilum [1]. Complete hepatic resection of localized tumors remains the only intent-to-cure treatment and requires either right- or left-sided hepatic resection [2].

PHC are most often classified according to the Bismuth-Corlette (BC) system [3]. Type IV PHC, characterized by involvement of both second-order biliary confluences are no longer regarded as a contraindication to resection. Indeed, Korean and Japanese teams showed that Type IV PHC could be resected but with high postoperative morbidity due to significant parenchymal sacrifice and vascular reconstructions of the branches of the hepatic artery and the portal vein [4,5]. 

Not all BC type IV can be operated on. Indeed, a left trisectionectomy (LTS) or right trisectionectomy (RTS) can only be performed when segments 6–7 and/or segments 2–3 can be preserved, respectively [6,7]. When both second-order biliary confluences are free, liver resection is naturally oriented towards the preservation of the left lateral section (segments 2–3) because an LTS is technically more difficult and more often requires reconstruction of the branches of the hepatic artery or portal vein [4,8,9]. 

A recent study of benchmark values for outcomes after PHC resection suggested a better overall survival associated with left-sided liver resections; however, this study did not discriminate between hemihepatectomies and trisectionectomies [10]. Similarly, Ebata et al. reported a large series of right and left trisectionectomies for BC type IV PHC but no comparison was made between RTS and LTS [4]. Thus, whether resection of BC type IV PHC necessitates LTS or RTS remains an unanswered question. The aim of the present study was to evaluate and compare the perioperative outcomes and long-term prognosis of right and left trisectionectomies in the treatment of type IV PHC. We present the following study in accordance with the STROBE reporting checklist.

## 2. Methods

All patients with suspected PHC referred to the Rennes’ University Hospital between January 2012 and December 2019 were included prospectively and analyzed retrospectively.

### 2.1. Ethical Assessment

All subjects consented to inclusion in the study. The study was conducted in accordance with the Declaration of Helsinki and the research protocol was reviewed and approved by the Research Ethics Committee of the University of Rennes 1 (Approval number: 20.142). All participants were informed and gave written consent for inclusion in the study and use of their medical information for scientific research. Data quality was ensured by prospective collection.

### 2.2. Preoperative Workup and Resectability Assessment

In this study, we referred to the terminology of the International Hepato-Pancreato-Biliary Association to describe liver anatomy [6] and we referred to the recently launched New World Classification to describe hepatic resections [11].

On presentation, patients underwent computed tomography (CT scan), magnetic resonance imaging and cholangiopancreatography (MRI/MRCP). Both the senior surgeon and the radiologist simultaneously analyzed the imaging. Lobar/segmental liver atrophy ipsilateal to the tumor was recorded and based on the following features: obvious reduction in liver lobe volume associated with portal vein thrombosis, presence of ductal or vessel crowding present on MRI/MRCP and lower attenuation of the atrophied liver parenchyma on pre-contrast computed tomography. Liver, peritoneal or lung metastases were contraindications to surgery. Candidates for surgery were sorted using BC classification (3, 8) and the Rennes XY classification system in order to assess the B2–B3 biliary confluence status (8). In BC type IV–Rennes X patients (the B2–B3 confluence was involved), a LTS (H123458-B) was indicated when the B6–B7 confluence was free. In BC type IV–Rennes Y patients (the B2–B3 confluence was free), resection was naturally oriented towards a RTS (H145678-B), even if the B6–7 confluence was free. 

Preoperative biliary drainage involved the future liver remnant (FLR). Percutaneous transhepatic biliary drainage (PTBD) was routinely performed until 2016. Thereafter, endoscopic biliary drainage (EBD) was the preferred method [12]. When EBD was not feasible, PTBD was performed instead. One week before surgery, patients with PTBD were admitted for bilio-enteric instillation, enteral nutrition and were operated on if a serum total bilirubin was ≤50 µmol/L [13]. The need for portal and arterial resection and reconstruction was estimated peroperatively. Total liver volumes (TLV) and future liver remnant volumes (FLRV) were measured in order to calculate the referral FLRV/TLV. When the referral FLRV/TLV rate was <35%, portal vein embolization ipsilateral to the future liver to be resected was performed. 

### 2.3. Surgical Technique

Liver resections were performed by two surgeons (KB and HJ). Peritoneal seeding, liver metastases, or tumor seeding around the biliary transhepatic drain of the remnant liver contraindicated resection. The presence of resectable metastatic nodes in the hilum or at the origin of the common hepatic artery did not contraindicate resection. All patients underwent right- or left-sided hepatic resection en bloc with S1, extrahepatic bile duct and extended lymphadenectomy of the hepatoduodenal ligament and coeliac region. Bilioenteric continuity was re-established by means of a Roux-en-Y hepaticojejunostomy. When vascular resections were necessary, arterial reconstruction was performed early in the procedure before liver parenchymal transection. The aim was to obtain a patent arterial anastomosis of the future liver remnant in order to minimize futile liver resections. Arterial reconstruction was performed as an end-to-end anastomosis between different branches of the hepatic artery as needed or with the use of a rotating splenic artery when arterial length (even after sectioning of the gastroduodenal artery) was not sufficient. If none of these techniques were feasible, an autologous greater saphenous graft was used. Portal vein resections were performed as the last step of the procedure and a direct end-to-end anastomosis using a “growth-factor” was always performed. When portal vein length was not sufficient, an autologous iliac, jugular or left renal vein graft was used. 

Postoperative follow-up included daily clinical examination, and biochemical and liver functional tests. A computed tomography (CT) scan was performed on demand depending on the postoperative course. Patients undergoing vascular resections underwent a systematic Doppler ultrasound confirmation of vascular permeability before hospital discharge or a CT angiography if the ultrasound was not contributory. 

### 2.4. Histological Analyses

Intraoperative frozen section assessment of the proximal and distal ductal margins was routinely performed. The longest tumor diameter, level of differentiation, involvement of the resected portal and arterial branches, and the presence or absence of microvascular or perineural invasion were all reported. Tumor vascular invasion was recorded as to whether it considered the resected liver or the future liver remnant. R1 resection was defined as the presence of tumor cells at the ductal, vascular or parenchymal margins based on the histology of the specimen. Tumors were staged according to the Eighth Edition of the Union for International Cancer Control (UICC) Classification of Malignant Tumours [14].

### 2.5. Adjuvant Chemotherapy

Until February 2014, gemcitabine + oxaliplatin were administrated for 6 months to all patients as part of the Prodige 12-Accord 18-Unicancer GI trial [15]. Thereafter, no postoperative chemotherapy was administered until the results of the BILCAP study were presented in 2019 [16]. All patients then received capecitabine for 6 months. 

### 2.6. Outcomes

The aim of our study was to compare the outcomes of RTS and LTS in type IV PHC. The following data were analyzed: the resectability rate, need for vascular resections, radicality rate, incidence of suspected vascular (arterial and portal) invasion and eventually, the histologically-proven invasion rate, postoperative complication rate within 90 days according to the Clavien–Dindo classification (CDC) with major complications defined as grades > IIIa, postoperative liver failure (POLF) rate defined according to the International Study Group of Liver Surgery [17], recurrence rate and mortality rate. Postoperative mortality was defined by the occurrence of death within 90 days of the operation or at any time during the postoperative hospital stay. Patients were followed up every 3 months for the first 2 years, every 6 months in the following 3 years and then yearly. Local or distant recurrence was detected with CT scan and MRI. Medical record consultation or direct contact with the participant assured there was no loss to follow-up. Overall (OS) and disease-free (DFS) survival were calculated and compared while the independent risk factors influencing OS and DFS were explored. 

### 2.7. Statistical Analysis

Continuous variables were expressed as medians with the interquartile range, and categorical variables were summarized as frequency and percentage. There were no missing data. Statistical analysis was performed using the Student’s *t*-test, χ^2^ test, Fisher exact probability test or Mann–Whitney U test where appropriate. 

For overall (OS) and disease-free (DFS) survival, cumulative event curves (censored endpoints) were generated using the Kaplan–Meier method. Patient survival was determined from the time of surgery to the time of death or most recent follow-up. Patients who died without recurrence were not censored. Differences in survival curves were compared using the log-rank test. Independent predictors of survival time were studied using the Cox proportional hazards model, and all pre-, per- and postoperative predictors with a *p* value < 0.1 were retained in the model. *p* values < 0.05 denoted statistically significant differences. Analysis was performed using the SPSS statistics software version 22.0.

## 3. Results

### 3.1. Study Population

From January 2012 to December 2019, 130 consecutive patients with suspected PHC were referred to our center. Seventeen patients were excluded from the study either due to initial metastatic dissemination (9 patients) or due to inclusion in the transplantation arm of a prospective randomized study comparing hepatic resection versus liver transplantation (ClinicalTrials.gov, NCT02232932) (8 patients). Out of 113 (87%) patients scheduled for radical resection, 102 (78.4%) patients were finally operated on as the rest of the patients presented a perioperative contraindication to resection (microscopic peritoneal metastasis, cirrhosis). Out of 102 resected patients, 67 (65%) presented a BC type IV PHC. Among the 67 patients, 25 (37.3%) patients presented a BC type IV–Rennes X PHC and underwent an LTS, while 42 (62.6%) patients presented a BC type IV–Rennes Y PHC and underwent an RTS.

### 3.2. Baseline Characteristics

In this study, all patients were Caucasians of French origin. The distribution of the patients’ baseline characteristics was similar in the LTS and RTS groups, notably when considering biliary drainage, preoperative bilirubin level and time interval from drainage to surgery (Table 1). 

Compared with the LTS group, the rate of portal vein embolization (PVE) was significantly higher in the RTS group (4% vs. 83.3%; *p* < 0.01). In the RTS group, the referral volume of the remnant left lateral section increased significantly after PVE (24 ± 2% to 32 ± 1%; *p* = 0.009). However, even after PVE, it remained smaller than the referral volume of the right lateral section in the LTS group (FLRV/TLV: 38% ± 1% vs. 32% ± 1%; *p* = 0.009) (Table 1). Interestingly, the decision to perform surgery was driven by jaundice resolution as the time interval from diagnosis to surgery was similar between groups (6 weeks [5,6,7,8,9] vs. 6 weeks [3,4,5,6,7,8,9,10], *p* = 0.33) even though portal vein embolization was almost exclusively used in the RTS group (Table 1).

### 3.3. Vascular Resections

The rate of portal vein resection was similar in the LTS (*n* = 10, 40%) and RTS (*n* = 22, 52.4%) groups (*p* = 0.29). In RTS, a direct end-to-end portal anastomosis was performed for all patients, while in LTS it was only feasible for eight patients. The two remaining patients had an autologous right external iliac vein interposition graft. Six arterial resections (24%) were performed in the LTS group while none were performed in the RTS group. Arterial reconstruction was performed as a direct end-to-end anastomosis between the right or right posterior section hepatic artery and common or proper hepatic artery with the use of 6/0 or 7/0 interrupted polypropylene sutures.

### 3.4. Perioperative Complications

Severe perioperative complications (Clavien–Dindo > IIIa), vascular complications as well as mortality rates within 90 days were similar in the LTS and RTS groups. The postoperative liver failure (POLF) rate was higher after RTS (38.1% vs. 16%; *p* = 0.048), and severe (Grade B/C) POLF was more frequent even though the latter did not reach statistical significance (26% vs. 8%, *p* = 0.06) (Table 2).

### 3.5. Histological Findings

The histological characteristics of the tumors are reported in Table 3. Tumor histologic differentiation, grade, and size as well as the rate of regional lymph node metastases were similar in the LTS and RTS groups. The rate of portal (36% vs. 26%, *p* = 0.22) and arterial invasion (16% vs. 4.8%, *p* = 0.33) was similar in the LTS and RTS groups. Finally, the R0 resection margin rate was higher in the LTS group compared with the RTS group (92% and 81%) although this difference did not reach statistical significance (*p* = 0.154).

### 3.6. Survival Analysis and Prognostic Factors

The median time of follow-up was 25 months (IQR: 8–42) while median survival for both groups (LTS and RTS) was 50 months and OS rates were 77%, 62% and 30% at 1, 3 and 5 years, respectively. The OS rate was significantly higher after LTS compared with RTS (92%, 89%, 66% vs. 66%, 49%, 30% at 1, 3 and 5 years, respectively; *p* = 0.009) (Figure 1A). Although this difference was also observed when disease-free survival was compared it did not reach statistical significance (91%, 69%, 43% vs. 90%, 43%, 18% at 1, 3 and 5 years, respectively; *p* = 0.110) (Figure 1B). 

Thirteen clinical and pathological variables were analyzed as potential prognostic factors of survival (Table 4). Multivariable analysis identified LTS (HR: 0.31, 95% CI: 0.13–0.75, *p* = 0.009), high histological differentiation (HR: 0.11, 95% CI: 0.15–0.83, *p* = 0.033) and portal microscopic vein invasion (HR: 3.5, 95% CI: 1.3–9.47, *p* < 0.001) as independent prognostic factors of OS, while radicality of resection (HR: 0.80, 95% CI: 0.30–0.96, *p* = 0.049) and the absence of lymph node invasion (HR: 0.76, 95% CI: 0.70–0.93, *p* = 0.037) were independent prognostic factors of DFS. 

## 4. Discussion

Margin-free resection remains the only curative-intent treatment for localized PHC and tumor extension along the right and/or left hepatic ducts determines the side and the extent of liver parenchymal resection [3,8,18]. In this study, we compared the results of RTS and LTS for Type IV BC PHC, where both second-order biliary confluences are involved, and we found that LTS, although technically more demanding and more frequently associated with arterial reconstructions, provided similar radicality and higher overall survival rates compared with RTS. Although few reports have compared right liver resections with left liver resections in PHC (including hemihepatectomies and trisectionectomies), to our knowledge, no other series have compared RTS with LTS [19,20,21,22,23]. Ebata et al. reported a large series of right and left trisectionectomies for BC type IV PHC but no comparison was made between RTS and LTS [4]. 

Since tumor localization determines which side of the liver requires resection, it could seem futile to compare RTS with LTS. Indeed, surgical technique choice is often limited by the extension and side of the tumor, the potential need for vascular reconstruction, and finally, the FLRV/TLV ratio, which dictates the need for ipsilateral portal vein embolization of the liver to be removed. In fact, when both secondary biliary confluences are free, the lower occurrence of arterial reconstructions in RTS compared with LTS and the possibility of increasing hepatic volume with portal vein embolization in the RTS orients surgeons towards performing an RTS. However, when it comes to type IV PHC where both secondary biliary confluences are outcropped by the tumor, the question is whether hepatic resection should be left- or right-sided. In these situations, the surgical strategy to resect a PHC is naturally oriented towards the preservation of the left lateral section, even if the right lateral section can be preserved. Indeed, the left lateral section of the liver represents an easily recognizable anatomical entity, the left hepatic duct is long and the left portal vein is absolutely constant [24]. Moreover, the left hepatic artery runs along the left edge of the hepatic pedicle and is therefore rarely involved, just as in the RTS group in our series [8,9,20]. LTS are more technically demanding than RTS since the intersectional limit of the right lateral section (segments 6, 7) is not apparent and the plane of the right lateral fissure is large [25]. Finally, owing to the close vicinity of the right hepatic artery and portal bifurcation with the biliary confluence, LTS may more frequently necessitate arterial and venous reconstructions [8,9]. 

In our study, while overall survival was significantly higher in the LTS compared with the RTS group, disease-free survival, although higher in the LTS group, did not reach statistical significance. This observation could be due to low statistical power related to the small number of disease-recurrence events in an already limited study population. Since tumor characteristics, interval from diagnosis to surgery, patterns of disease recurrence (local or distal) and chemotherapy regimens did not differ between the RTS and LTS groups, we hypothesize that this difference may be multifactorial and could also be related to a higher rate of severe postoperative liver failure (Grade B/C) in the RTS group (26% vs. 8%, *p* = 0.06). Postoperative liver failure has been shown to be associated with early postoperative mortality but also lower long-term survival [26,27]. Indeed, analysis of the RTS group survival pattern shows that nearly 50% of deaths occurred within the first year after surgery. This hypothesis was also proposed in the recently published benchmark study on PHC by Mueller et al., which showed a 90-day mortality rate that was four times as high in right-sided vs. left-sided hepatectomies (7.3% vs. 1.8%) together with a significantly inferior overall survival (45 vs. 61 months, *p* = 0.002), mostly related to a higher rate of liver failure in case of right-sided hepatectomies [10]. 

In this study, the higher OS in the LTS group could be also related to the slightly higher rate of R0 resection (92% vs. 81%) even though, due to low statistical power related to limited number of patients analyzed, this difference did not reach statistical significance. This unexpectedly high overall survival rate of patients undergoing LTS in our series suggests that our therapeutic strategy in type IV PHC resection may need to be revisited. Indeed, when the biliary duct of the right lateral section, i.e., the confluence of B6–B7, is free, it may be preferable to consider performing LTS even if the left lateral section can be preserved or/and the right hepatic artery appears to be involved by the tumor. 

Our results also suggest that the need to reconstruct the hepatic artery or its branches, which is a prerogative of the left-sided hepatic resections, should no longer be an obstacle to surgery. Current advances in complex liver and vascular surgical techniques have led to more aggressive approaches with acceptable outcomes, low complication rates and substantial survival benefits compared with non-surgical strategies [9,28,29,30]. Even though the type of vascular reconstruction (portal, arterial or both) may influence postoperative outcomes it appears beneficial in terms of long-term survival as recently reported by Angelico et al. in a recent systematic review [31]. These encouraging results has led to an increase in the popularity of LTS, especially among Asian surgical teams. Mizuno et al. recently reported the results of the largest series of PHC resections associated with hepatic artery reconstruction, alone or combined with portal vein reconstruction. The majority (93%) of artery reconstructions were associated with left-sided hepatic resection and more frequently with LTS (59%) [9]. Our study is also in line with these results since all arterial resections were performed in the LTS group. 

The limitations of this comparative study are its single-center design, associated with a possible selection bias related to the surgeon’s experience in choosing one technique over the other (RTS or LTS), its retrospective nature, the short follow-up, and finally, the relatively small number of reported cases, which is especially true for LTS (25 patients) compared with RTS (42 patients). However, PHC is a rare disease and analysis of only type IV PHC can be challenging as it is a highly selected population. A randomized controlled trial comparing the different types of hepatic resection is unlikely to be undertaken given the very low disease incidence in western countries [32]. Nevertheless, we are currently initiating a nationwide multicenter French retrospective study of all prospectively collected data on PHC.

Finally, our study is the first to compare surgical and survival outcomes of LTS with RTS and we believe that our results are likely to challenge the manner in which the surgical management of type IV PHC is commonly implemented. Indeed, rather than evaluating resectability in a “hepatofugal” way, i.e., starting from the tumor and evaluating its extension in the intrahepatic bile ducts, we think that reversing the surgical approach, i.e., evaluating resectability in a centripetal way by analyzing the biliary ducts remaining free from invasion and favoring preservation of the right lateral sector even at the cost of arterial or venous resection.

## 5. Conclusions

In conclusion, left trisectionectomies may play an important role and provide substantial survival benefits in type IV PHC, even if arterial reconstructions are more often necessary. Whenever the right posterior biliary duct confluence can be preserved, a left trisectionectomy may be considered even if the B2–B3 biliary confluence remains tumor-free.

## Figures and Tables

**Figure 1 cancers-14-02791-f001:**
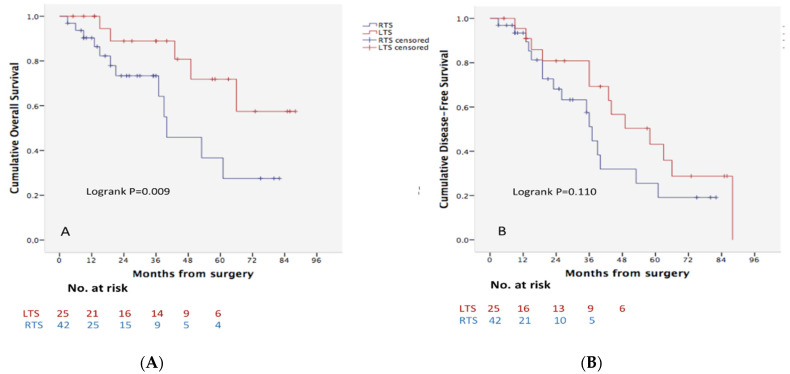
Kaplan–Meier survival analyses. Overall (**A**) and disease-free (**B**) survival curves according to trisectionectomy side. LTS: Left Trisectionectomy; RTS: Right Trisectionectomy.

**Table 1 cancers-14-02791-t001:** Preoperative characteristics and management of patients undergoing RTS or LTS.

Patient Characteristics	LTS (*n* = 25)	RTS (*n* = 42)	*p*-Value
Age, years, median [IQR]	65.5 (56–76)	70 (62–73)	0.25
Male gender, *n* (%)	16 (64)	41 (40.1)	0.19
BMI, kg/m^2^, median [IQR]	25 (20–26)	23 (21–26)	0.66
ASA score ≤ 2, *n* (%)	18 (72)	33 (78.6)	0.37
Jaundice at referral, *n* (%)	14 (44)	19 (54)	0.57
Bilirubin at referral, µmol/L, median [IQR]	194 (28–331)	277 (120–402)	0.12
Bilirubin at surgery, µmol/L, median [IQR]	46 (14–102)	40 (26–73)	0.60
Preoperative biliary drainage, *n* (%)	19 (76)	39 (92.9)	0.13
None, *n*	6	3	
PTBD ^1^, *n*	11	25	
EBD, *n*	8	14	
Bilio-enteric instillation	9 (36)	21 (50)	0.19
Hemi liver atrophy ^2^, *n* (%)	2 (8)	4 (10)	0.83
Portal vein embolization, *n* (%)	1 (4) ^3^	35 (83.3) ^4^	<0.01
Time from diagnosis to surgery, weeks, median [IQR]	6 (5–9)	6 (3–10)	0.33
FLRV/TLV (%) at referral, mean ± SD	38 ± 1	24 ± 2	0.009
FLRV/TLV (%) at surgery, mean ± SD	38 ± 1 ^5^	32 ± 1	0.01

LTS: Left Trisectionectomy; RTS: Right Trisectionectomy; PHC: Perihilar Cholangiocarcinoma; BMI = Body Mass Index; PTBD = Percutaneous Transhepatic Biliary Drainage; EBD = Endoscopic Biliary Drainage; FLRV = Future Liver Remnant Volume; TLV = Total Liver Volume; SD: standard deviation. ^1^ Upfront or after EBD failure; ^2^ Atrophy of the future resected liver; ^3^ Embolization of the left and right anterior portal branches; ^4^ Embolization of the right portal branch and/or right/middle hepatic vein; ^5^ Only one patient had embolization of the left and right anterior portal branches. FLRV volumes were not recalculated prior to surgery.

**Table 2 cancers-14-02791-t002:** Perioperative (90 days) surgical complications.

Complications	LTS (*n* = 25)	RTS (*n* = 42)	*p*-Value
Clavien–Dindo IIIb and IV, *n* (%)	7 (30)	4 (12.5)	0.10
Vascular complications, *n* (%)	3 (12)	4 (9.3)	0.38
Biliary Fistula, Grade B/C, *n* (%)	9 (36)	8 (19)	0.50
POLF, *n* (%)	4 (16)	16 (38.1)	0.04
POLF Grade B/C, *n* (%)	2 (8)	11 (26)	0.06
Deaths, *n* (%)	2 (8)	5 (11.9)	0.13

LTS: Left Trisectionectomy; RTS: Right Trisectionectomy; POLF: Postoperative liver failure.

**Table 3 cancers-14-02791-t003:** Histological characteristics of the specimen.

Histological Characteristics	LTS (*n* = 25)	RTS (*n* = 42)	*p*-Value
Tumor diameter, max (mm) (IQR)	25 (21–37)	25 (22–32)	0.90
Harvested lymph nodes, *n* (%)	5 (3–8)	6 (4–8)	0.90
TNM classification (UICC 8th) *n* (%)			0.35
pT1	12 (48.0)	27 (64.3)	
pT2	10 (40.0)	13 (30.9)	
pT3	3 (12.0)	1 (2.3)	
pT4	0	1 (2.3)	
N classification, *n* (%)			0.44
pN1/2	8 (32.0)	12 (28.5)	
M classification, *n* (%)			0.13
pM1	2 (8.0)	0	
Invaded lymph nodes, *n* (%)	8 (33.3)	12 (28.6)	0.52
R1 resection, *n* (%)	2 (8)	8 (19)	0.15
Portal vein invasion, *n* (%) *	9 (36)	11 (26)	0.22
Arterial invasion, *n* (%) *	4 (16)	2 (4.8)	0.33
Perineural invasion, *n* (%)	21 (87.5)	36 (85.7)	0.10
Tumor Grade: Moderate/Low differentiation, *n* (%)	9 (36)	14 (33.3)	0.54

* Tumor vascular invasion may involve both the resected and future liver remnant and is independent of arterial and portal resections; LTS: Left Trisectionectomy; RTS: Right Trisectionectomy.

**Table 4 cancers-14-02791-t004:** Predictors of overall survival and recurrence-free survival.

Analysis	Overall Survival			Disease-Free Survival		
	Univariable	Multivariable		Univariable	Multivariable	
Variable	*p*-Value	Hazard Ratio (95% CI)	*p*-Value	*p*-Value	Hazard Ratio (95% CI)	*p*-Value
Age (<67 years)	0.004	0.30 (0.12–0.60)	0.190	0.3	-	-
Sex	0.2	-	-	0.3	-	-
ASA Score (<3)	0.9	-	-	0.7	-	-
Bilirubin level (>50 μmol/L)	0.3	-	-	1.0	-	-
Postoperative liver failure	0.3	-	-	0.7	-	-
Radicality of resection (R0)	0.006	0.70 (0.50–0.95)	0.091	0.003	0.80 (0.30–0.96)	0.049
Tumor size (<25 mm)	0.6	-	-	0.2	-	-
Lymph node invasion (N0)	0.3	-	-	0.004	0.76 (0.70–0.93)	0.037
Histologic differentiation	<0.001	-	0.05	0.2	-	-
High	-	0.11 (0.15–0.83)	0.033	-	-	-
Moderate/Low	-	0.13 (0.33–1.40)	0.113	-	-	-
Perineural Invasion	0.6	-	-	0.7	-	-
Hepatic artery Invasion	1.0	-	-	0.5	-	-
Portal vein Invasion	0.02	3.5 (1.30–9.47)	<0.001	0.2	-	-
Type of liver resection	0.013	-	0.009	0.3	-	-
RTS	-	1.00	-	-	-	-
LTS		0.31 (0.13–0.75)	-	-	-	-

RTS = Right Trisectionectomy; LTS = Left Trisectionectomy.

## Data Availability

The data presented in this study are available on request from the corresponding author. The data are not publicly available due to confidentiality.

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
