# Peer review of "Liver Resection for Type IV Perihilar Cholangiocarcinoma: Left or Right Trisectionectomy?"

_cancers, 2022, doi:10.3390/cancers14112791_

Round 1

Reviewer 1 Report

The article is very original. The orientation and predilection of the Authors toward a left trisectionectomy when both second order biliary confluences are involved, might represent a "surgical hazard" due to higher complexity and higher rate of vascular resections, particularly of the arterial side but it might be worthy in relation to a higher overall survival.

In addition, even if not statistically significant, the Authors noticed a higher disease free survival and R0 resection rates of the left trisectionectomy compared to the right trisectionectomy.

Author Response

Manuscript ID: Cancers-1730112

Please find attached our response to all the issues raised. We would like to thank the reviewer for his/her helpful and constructive comments. We have considered his/her suggestions and we think that the changes they suggested contributed to substantially improve the quality of this manuscript.

Reviewer 1: comments to the author

The article is very original. The orientation and predilection of the Authors toward a left trisectionectomy when both second order biliary confluences are involved, might represent a "surgical hazard" due to higher complexity and higher rate of vascular resections, particularly of the arterial side but it might be worthy in relation to a higher overall survival. In addition, even if not statistically significant, the Authors noticed a higher disease free survival and R0 resection rates of the left trisectionectomy compared to the right trisectionectomy.

Response: We thank the reviewer for his/her comments and encouraging remarks.

Indeed, we were surprised to find that despite a greater technical difficulty, left trisectionectomies were associated with a better overall survival than right trisectionectomies in type IV PHC. The explanation that we wanted to share with the reader is that if a left extended hepatectomy is technically more demanding, the volume and the anatomical layout of the remnant right posterior sector may be more favorable to straightforward postoperative course and long-term outcomes. To follow the reviewer's advice, our manuscript has been fully proofread by a native English editor. We thank again the reviewer for understanding the puzzling concept that we wanted to put forward.

Reviewer 2 Report

The aim of the study was to compare the right trisegmentectomy (DTS) to the left trisegmentectomy (LTS), showing an higher overall survival of LTS.

The article is well written and the topic interesting, but the authors should note these comments:

-in the discussion, a note should be made on the small group of patients who have been analyzed in particular the small number of patients doing LTS (25) compared to RTS (42)

- It isn’t clear the ethnic origin of the patients, knowing the higher incidence in Eastern countries. We must consider the Japanese and Korean studies that explain how, after a major resection of the liver, we have a high morbidity due to parenchymal sacrifice.

- The choice of RTS when both second biliary bile ducts confluence are free, the lower recurrence of arterial and venous reconstruction in RTS rather than LTS, the possibility of increasing hepatic volume with portal vein embolization in the RTS, presupposes a preference in choice this method and should be discussed by the authors

- The limited possibility of choosing the surgical technique in case of non-engagement of the 6-7 segment (RTS) or the 2-3 (LTS) or of the extension and side of the tumor that forces us to choose one technique over the other and a FLRV / TLV that leads us to necessarily prefer the embolization of the portal vein ipsilateral to the future liver removed

-The involvement of vascular structure is crucial for the surgical technique. Description of vascular reconstruction should be reported. In the discussion, the importance of the type of vascular reconstruction should be included because it influences outcomes as recently reported by a systematic review which should be mentioned in the discussion [Vascular Involvements in Cholangiocarcinoma: Tips and Tricks. Cancers (Basel). 2021 Jul 25;13(15):3735. doi: 10.3390/cancers13153735]

-An important limit, in addition to the single center study, is the experience of the surgeon who may be more inclined to one technique than the other. Also, the difference in the chemotherapy regimen used up to 2014 (gemcitabine + oxaliplatin) compared to the lack of post-operative chemotherapy after February 2014, in this case there are no written or analyzed different survivals or any differences in mortality or tumor recurrence.

-Finally, the lack of description of the tests used in the follow up, where there could be evidence of a possible alteration of the portal vein or hepatic artery flows should be commented. Also, a median follow-up of 25 months may not be sufficient.

Author Response

Manuscript ID: Cancers-1730112

Please find attached our response to all the issues raised. We would like to thank the reviewer for his/her helpful and constructive comments. We have considered his/her suggestions and we think that the changes they suggested contributed to substantially improve the quality of this manuscript.

Reviewer 2: comments to the author

The aim of the study was to compare the right trisegmentectomy (DTS) to the left trisegmentectomy (LTS), showing a higher overall survival of LTS.

The article is well written and the topic interesting, but the authors should note these comments:

  1. In the discussion, a note should be made on the small group of patients who have been analyzed in particular the small number of patients doing LTS (25) compared to RTS (42).

Response: We thank the reviewer for his/her comment.  Appropriate changes have been made in the manuscript. A sentence reporting this remark has been added in the Discussion section (See paragraph 6, line 3-5, page 10) ‘…. and finally the relatively small number of reported cases, especially true for LTS (25 patients) compared with RTS (42 patients)).

  1. It isn’t clear the ethnic origin of the patients, knowing the higher incidence in Eastern countries. We must consider the Japanese and Korean studies that explain how, after a major resection of the liver, we have a high morbidity due to parenchymal sacrifice.

Response: We thank the reviewer for his/her insightful comment. In this series all patients were French and of Caucasian origin.

As it is well noticed by the reviewer, parenchymal sacrifice may lead to high postoperative morbidity and mortality rate. Indeed, as reported in Asian cholangiocarcinoma studies, high perioperative morbidity was also observed in our series. The rate of liver failure reached up to 38% in case of RTS, biliary fistula rate reached 36% in case of LTS and postoperative death was 8% for LTS and 11.9% for RTS (Table 2).

Appropriate changes have been made in the manuscript. The information about ethnic origin of patients was added in the Results section (See 3.2 Baseline characteristics, line 1, page 6) and is highlighted in red.

  1. The choice of RTS when both second biliary bile ducts confluence are free, the lower recurrence of arterial and venous reconstruction in RTS rather than LTS, the possibility of increasing hepatic volume with portal vein embolization in the RTS, presupposes a preference in choice this method and should be discussed by the authors

Response: We fully agree with this comment. Indeed naturally surgeons are oriented towards performing a RTS in order to avoid vascular reconstructions. It is also the case in our experience and this is the reason why LTS (25 patients) were less frequently performed compared with RTS (42 patients). This point has been added in the Discussion section (See paragraph 2, lines 5-8, page 9).

  1. The limited possibility of choosing the surgical technique in case of non-engagement of the 6-7 segment (RTS) or the 2-3 (LTS) or of the extension and side of the tumor that forces us to choose one technique over the other and a FLRV / TLV that leads us to necessarily prefer the embolization of the portal vein ipsilateral to the future liver removed

Response: We thank the reviewer for this constructive suggestion. Indeed tumor side extension, vascular involvement and FLRV / TLV ratio dictate the surgical technique. However, the message we wanted to convey is that in certain cases when the pedicle of the right lateral section is free, it may be preferable to preserve the right lateral section, even if a RTS is feasible and easier to perform. This message is puzzling and constitutes the originality of our study.  To follow the reviewer recommendation, we have summarized the principles of PHC resection in the Discussion section (See paragraph 2, lines 2-5, page 9).

  1. The involvement of vascular structure is crucial for the surgical technique. Description of vascular reconstruction should be reported. In the discussion, the importance of the type of vascular reconstruction should be included because it influences outcomes as recently reported by a systematic review which should be mentioned in the discussion [Vascular Involvements in Cholangiocarcinoma: Tips and Tricks. Cancers (Basel). 2021 Jul 25;13(15):3735. doi: 10.3390/cancers13153735]

Response: We thank the reviewer for his insightful suggestion. Liver resection and vascular reconstruction techniques were not the main subject of this study but we strongly agree that the type of vascular reconstruction is important in terms of short postoperative outcomes. A brief summary of our surgical vascular technique was added in the Methods section. (See 2.3 Surgical technique, line 8-17, page 3-4).

Moreover, description of the vascular reconstructions performed in our population was added in the Results section (See 3.3 Vascular resections, line 2-8, page 6).

Finally, the importance of the type of vascular reconstruction was included in the Discussion section (See paragraph 5, line 6-9, page 10) and the systematic review of Angelico et al. was mentioned and referenced (Reference number: 31). All changes have been highlighted in red.

  1. An important limit, in addition to the single center study, is the experience of the surgeon who may be more inclined to one technique than the other. Also, the difference in the chemotherapy regimen used up to 2014 (gemcitabine + oxaliplatin) compared to the lack of post-operative chemotherapy after February 2014, in this case there are no written or analyzed different survivals or any differences in mortality or tumor recurrence.

Response: We thank the reviewer for his/her constructive comment. Indeed, surgeon’s experience could serve as a potential selection bias. Nevertheless, 2 senior surgeons performed all procedures and the choice of technique was decided after discussion between both of them before the procedure. Appropriate changes have been made and this comment was added in the Discussion section (See paragraph 6, lines 2-3, page 10).

As far as adjuvant chemotherapy is concerned, no separate survival analysis was performed, as current literature does not clearly support survival benefit for postoperative treatments. Indeed, neither Gemcitabine/Oxaliplatine nor Capecitabine offered a clear survival benefit in an intention-to-treat population. Moreover, in our study chemotherapy regimens followed were independent of surgical resection technique or histopathologic characteristics of PHC (they only depended on current national chemotherapy recommendations) and thus it is little likely that patients received different chemotherapy treatments.

  1. Finally, the lack of description of the tests used in the follow up, where there could be evidence of a possible alteration of the portal vein or hepatic artery flows should be commented. Also, a median follow-up of 25 months may not be sufficient

Response: We thank the reviewer for his comment that helps improving the quality of this study. As suggested, a brief summary of postoperative follow-up was added in the Methods section (See 2.3. Surgical technique, line 19-22, page 4).

A median follow-up of 25 months is indeed short but given the median overall survival of 40 months in resected PHC, this follow-up may be of interest. The limiting factor of a median follow-up of 25 months certainly limits the skope of our study and this was added in the Discussion section (See paragraph 6, line 3, page 10).

Round 2

Reviewer 2 Report

 The description is complete and all the correction are well insert in the text!